# GAOKAO-Eval: Does high scores truly reflect strong capabilities in LLMs?

## Abstract

Large Language Models (LLMs) are commonly evaluated using human-crafted benchmarks, under the premise that higher scores implicitly reflect stronger human-like performance. However, there is growing concern that LLMs may *"game"* these benchmarks due to data leakage, achieving high scores while struggling with tasks simple for humans. To substantively address the problem, we create **GAOKAO-Eval**, a comprehensive benchmark based on China's National College Entrance Examination (Gaokao), and conduct "closed-book" evaluations for representative models released prior to Gaokao. Contrary to prevailing consensus, even after addressing data leakage and comprehensiveness, GAOKAO-Eval reveals that high scores still fail to truly reflect human-aligned capabilities. To better understand this mismatch, We introduce the Rasch model from cognitive psychology to analyze LLM scoring patterns and identify two key discrepancies: 1) anomalous consistant performance across various question difficultiess, and 2) high variance in performance on questions of similar difficulty. In addition, we identified inconsistent grading of LLM-generated answers among teachers and recurring mistake patterns. we find that the phenomenon are well-grounded in the motivations behind OpenAI o1, and o1's reasoning-as-difficulties can mitigate the mismatch. These results show that GAOKAO-Eval can reveal limitations in LLM capabilities not captured by current benchmarks and highlight the need for more LLM-aligned difficulty analysis.

## 1 Introduction

Human-aligned capabilities, typically designed based on difficulty levels aligned with human performance, have been widely used to evaluate LLMs and steer further research initiatives (Lu et al., 2022a; Shi et al., 2024; Hendrycks et al., 2021b; Zhang et al., 2023). Implicit in this approach is the assumption that high scores on these benchmarks indicate human-aligned capabilities. However, there is a growing concern within the community that LLMs may be "gaming" these benchmarks——achieving high scores while demonstrating instability and unreliability when confronted with tasks that are simple for humans (Zhou et al., 2024). As shown in Figure 1, while LLMs may excel at complex questions, they often struggle with simpler ones. This inconsistency further indicates that an LLMs' high score of 90% does not necessarily reflect its ability to handle tasks that are considerably easier for humans, who typically score only 60%. Such findings raise a critical question: *Do high scores truly reflect human-aligned capabilities in LLMs?*

The community initially attributed the observed mismatch between benchmark performance and LLM capabilities to data leakage (Ni et al., 2024; Zhou et al., 2023) or the insufficient coverage of benchmarks in comprehensively evaluating specific skills. However, they have not addressed the critical discrepancies that persist in LLM performance. Recent studies,

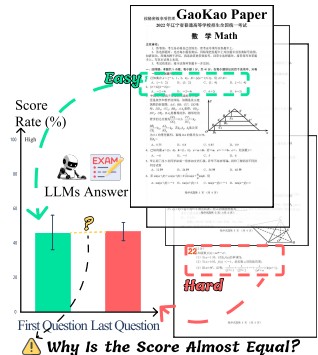

Figure 1: Comparison of LLM scores on the first and last questions of Gaokao Paper. Despite the latter being more difficult, LLMs achieve similar scores, revealing potential inconsistencies.

particularly in programming (Tambon et al., 2024) and reasoning tasks (Qiao et al., 2024), have observed intriguing patterns regarding the divergence between benchmark difficulty and Large Language Model (LLM) performance (Zhou et al., 2024). Nonetheless, these approaches mainly focus on specific problem types and cannot guarantee the absence of data leakage.

To substantively address the problem, we introduce GAOKAO-Eval, a comprehensive and annually updated benchmark based on Gaokao. The comprehensiveness and security of Gaokao is attributed to the process where Gaokao experts spend two months in a fully sealed environment each year, crafting 490 new questions that cover 2-3 key concepts from a pool of over 10,000 (Hubert et al., 2022). These questions are thoroughly tested with recruited students to ensure that the resulting scores follow a normal distribution. This rigorous process, combined with the exam's broad coverage of subjects and question types, makes Gaokao an ideal foundation for evaluating LLMs (Zong & Qiu, 2024; Zhang et al., 2023). Our framework evaluates only models released before the exam date, and employs over 54 high school teachers for grading subjective questions.

Through the rigorous evaluation process outlined above (see Figure 2), we uncovered a crucial insight: even after mitigating issues such as data leakage and insufficient benchmark coverage, an inherent conflict between LLM capabilities and benchmark design persists, as LLMs continue to exhibit inconsistent performance. Specifically, high scores do not necessarily reflect human-aligned capabilities in these models. Further analysis employs the theoretical human performance curve from cognitive psychology, modeled by the Rasch model, to rigorously characterize the deviation of LLM scoring patterns from human performance (Rasch, 1993; Bond & Fox, 2007). This reveals two statistical phenomena: a semi-difficulty-invariant scoring rate and high variance in performance on similarly difficult questions. We evaluate abundant representative models on GAOKAO-Eval in extensive scenarios to further investigate these phenomena, yielding several key takeaways: (1) grading inconsistencies compared to human examinees; (2) recurring error patterns across various task types; (3) the mitigation of the mismatch through o1's reasoning-as-difficulties approach.

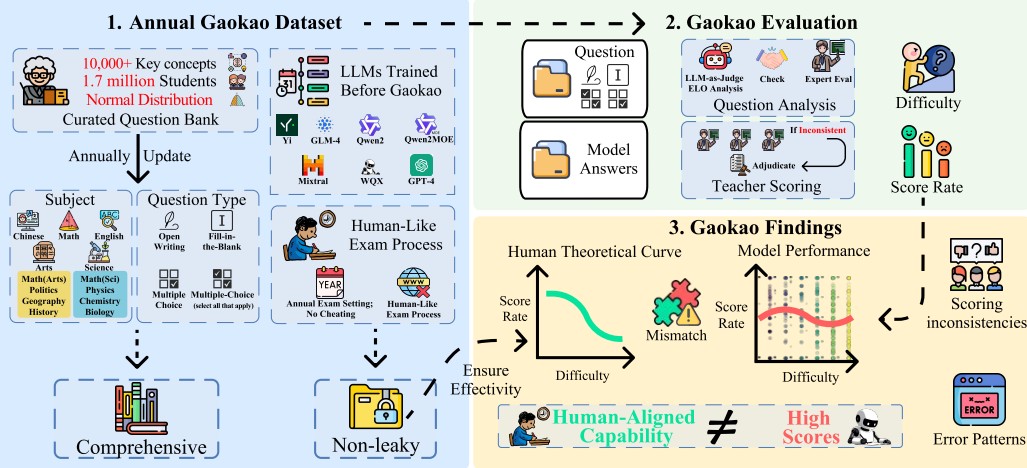

Figure 2: The GAOKAO-Eval pipeline. Built on the Gaokao benchmark, which ensures balanced difficulty and subject coverage, GAOKAO-Eval evaluates models released before the exam date under strict closed-book conditions, with human teachers grading subjective responses. Findings show that, even with high scores, LLMs have inconsistent scoring patterns and greater variation on tasks of similar difficulty. In contrast, human performance changes more predictably with task difficulty.

In summary, we make the following three-fold contributions:

- We introduce GAOKAO-Eval, a comprehensive and annually updated benchmark based on Gaokao. This benchmark provides a non-leaking, comprehensive assessment that closely mimics human-centric evaluation, encompassing a diverse range of subjects and question types.

- We reveal that high scores on benchmarks do not necessarily reflect human-aligned capabilities in LLMs. Our analysis shows that LLMs' scoring patterns deviate significantly from human performance, exhibiting semi difficulty-invariant distributions and high variance within similar difficulty levels.

- We identify that the mismatch is related to distinct error patterns and grading inconsistencies in LLMs compared to human examinees. Furthermore, we propose using reasoning tokens as a proxy for aligning LLM difficulties, which mitigates these issues.

## 2 GAOKAO-EVAL

GAOKAO-Eval leverages China's National College Entrance Examination as a comprehensive benchmark for evaluating LLMs. This evaluation framework is designed to provide a thorough assessment of LLM capabilities across various subjects and question types, while ensuring the integrity and relevance of the evaluation process. The benchmark is characterized by four key features: (1) full-paper examination covering all question types and subjects, including multimodal questions; (2) pre-exam open source evaluation, limiting assessment to models released before the current year's Gaokao; (3) expert grading by experienced Gaokao examiners for subjective questions; and (4) full transparency with open-source code, model responses, and scoring results.

Table 1: Comparison between **OURS** and other benchmarks. AU: Annual Update Size; Avg Q. Leng.: Avg Question Length; Expl.: Explanation; FB: Fill-in-the-Blank; MC: Multiple-Choice; BC: Binary-Choice; T. Chi.: Translated Chinese; N. Chi.: Native Chinese; S&A: Strict Security Measures and Annual Update; I. Type: Image Type.

| Benchmark | Size | AU | Avg. Q. | Expl. | Question | I. Type | Lang. | S&A |
|---|---|---|---|---|---|---|---|---|
| IconQA | 107K | - | 8.30 | ✗ | MC+FB | 1 | Eng. | ✗ |
| OK-VQA | 14K | - | 8.09 | ✗ | Open | 1 | Eng. | ✗ |
| Ai2D | 5K | - | 9.78 | ✗ | MC | 1 | Eng. | ✗ |
| FigureQA | 1M | - | 6.07 | ✗ | BC | 5 | Eng. | ✗ |
| ScienceQA | 6K | - | 12.11 | ✓ | MC | 5 | Eng. | ✗ |
| MMMU | 11.5K | - | 59.33 | ✓ | MC+Open | 30 | Eng. | ✗ |
| MMLU | 15K | - | 274.54 | ✗ | MC | 0 | Eng. | ✗ |
| MMLU Pro | 12K | - | 264.76 | ✓ | MC | 0 | Eng. | ✗ |
| CMMLU | 11K | - | 36.85 | ✗ | MC | 0 | N. Chi. | ✗ |
| C-Eval | 13K | - | 53.20 | ✓ | MC+BC | 0 | N. Chi. | ✗ |
| MM-Bench-CN | 3K | - | 15.48 | ✗ | MC | 20 | T. Chi. | ✗ |
| GAOKAO-MM | 0.65K | - | 260.19 | ✓ | MC | 32 | N. Chi. | ✗ |
| **OURS** | **3.95k** | **0.49k** | **674.01** | ✓ | MC+FB BC+Open | **32** | **N. Chi. + Eng.** | ✓ |

GAOKAO-Eval distinguishes itself from existing knowledge-based benchmarks through its comprehensive coverage, longer average question length, and employment of native Chinese data. Unlike previous benchmarks (Lu et al., 2022b; Marino et al., 2019; Kembhavi et al., 2016; Kahou et al., 2018; Lu et al., 2022a; Yue et al., 2024; Hendrycks et al., 2021b; Wang et al., 2024; Li et al., 2023; Huang et al., 2023; Liu et al., 2024), GAOKAO-Eval is based on the Gaokao-Bench (Zhang et al., 2023) and GAOKAO-MM (Zong & Qiu, 2024) benchmarks and features annual updates and human-scored evaluations, ensuring a secure and transparent framework for assessing LLMs.

### 2.1 COMPREHENSIVE GAOKAO-EVAL DESIGN

GAOKAO-Eval is meticulously designed to provide a thorough evaluation by encompassing multiple subjects, diverse question types, and various paper formats. This comprehensive approach ensures that LLMs are assessed across a wide range of cognitive tasks and knowledge domains.

**Subject and Question Type Coverage.** Figure 3a presents the distribution of question types across various subjects in GAOKAO-Eval. It encompasses not only multiple-choice questions but also more complex types such as short answers, essays, and subject-specific formats, providing

a thorough assessment of AI models' capabilities across different cognitive tasks and knowledge domains.

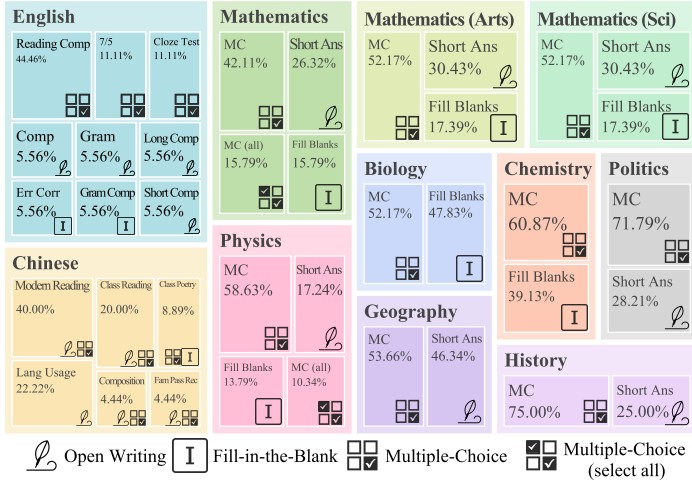 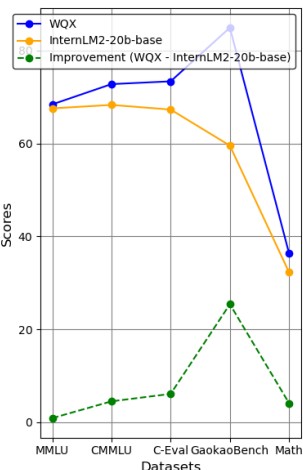

(a) Distribution of Question Types Across Subjects in GAOKAO-Eval.

(b) Performance Improvement: InternLM2-20b-base vs WQX.

Figure 3: Comprehensiveness of GAOKAO-Eval.

**LLM-Aligned Capabilities Improvement via GAOKAO Question.** Training on a specialized Gaokao dataset contributes to a broader enhancement of its capabilities, specifically reflected in its performance on other sophisticated knowledge benchmarks. In this motivation, we propose a specific model named WQX based on InternLM2-20b-base (Cai et al., 2024). The training detail about WQX is introduced in Appendix Section A. We evaluate the performance of WQX on several latest sophisticated knowledge benchmarks Math, MMLU, CMMLU, C-Eval, GaokaoBench. With an exceptional 84.94% accuracy on the GaokaoBench, the model demonstrates its superior capability in handling complex examination-style questions, a testament to the effectiveness of our targeted training and data augmentation strategies. The improvements observed in Math, MMLU, CMMLU, and C-Eval benchmarks further affirm the WQX model's comprehensive natural language understanding and its adeptness at navigating a wide array of knowledge-intensive tasks (see Figure 3b).The model's ability to better navigate these diverse and complex tasks suggests that the Gaokao dataset covers a broad spectrum of knowledge areas and cognitive skills.

## 2.2 EVALUATION METHODOLOGY AND SECURITY MEASURES

GAOKAO-Eval employs a rigorous evaluation methodology with strict security measures to ensure accurate, meaningful results while maintaining the integrity of the assessment process.

**Non-leaky Data and Temporal Isolation.** To address limitations in previous benchmarks like GAOKAO-Bench (Zhang et al., 2023) and GAOKAO-MM (Zong & Qiu, 2024), which potentially allowed data leakage, GAOKAO-Eval uses genuinely unseen data. It evaluates only open-source models released before June 6, 2024, ensuring temporal isolation and a closed-book environment. This approach provides a more objective assessment of LLMs' capabilities, avoiding the "open-book test" scenario present in earlier evaluations.

**Multimodal Evaluation Adaptations.** For multimodal questions, evaluation methods were adapted based on model capabilities. The Mixtral series, being language-only models, used only text input for multimodal questions. Due to poor performance of QwenVL-7B (Bai et al., 2023) on certain subjects, Qwen2-72B (Yang et al., 2024) text model was also evaluated on multimodal questions in Physics, Chemistry, and Geography for both New Curriculum Standard and National A test papers.

**Human Expert Grading.** 54 experienced Gaokao examiners graded the responses without prior knowledge of their AI origin. Clear guidelines were provided for handling misunderstood questions,

repeated answers, or explanations instead of direct answers. The evaluation considered the potential 1-2 point deviation in Chinese essays due to the lack of handwriting assessment, typically part of the Gaokao scoring process.

## 2.3 DATA PROCESSING

At the moment when the Gaokao concluded, we utilized various online channels to collect examination papers from various subjects. These papers were then standardized into a consistent format containing both text and images through manual processing. Specific processing techniques were applied according to the needs of each subject. For subjects such as Chinese, Mathematics, and English, which are commonly used across multiple provinces and tend to have fewer image-based questions, we opted to exclude any images. Consequently, we used solely textual questions as the data for evaluation. For subjects other than the aforementioned three, we treated each major question as a separate input. Sub-questions within each major question were formatted using (1), (2), and so on. Additionally, images within these questions were incorporated by employing the token to indicate their placement within the text. In scientific subjects, where formulas are prevalent, all mathematical expressions were converted into LaTeX format and encapsulated using the $ symbol, which ensured consistent and accurate representation of complex equations. To prevent any external instructions from influencing the model's performance, no extra prompts were included beyond the necessary formatting and data processing steps outlined above.

## 2.4 MODELS

The models' detailed information is listed in Appendix B.5. The models include Qwen2-72B (Yang et al., 2024) and Owen1-VL (Bai et al., 2023), Yi-34B (AI et al., 2024), GLM-4 (GLM et al., 2024), WQX, Mixtral (Mistral AI, 2024), and GPT-4o (OpenAI et al., 2024). Given the prevalence of graphical elements in high school examination questions, LLMs tend to respond only to text-based questions (with few exceptions), whereas multimodal LLMs address all types of questions.

## 3 RESULTS AND KEY FINDINGS

The following sections break down our key findings, focusing on the performance discrepancies between human-aligned capabilities and LLM-generated results. We systematically explore performance, difficulty ratings, unique error patterns exhibited by LLMs, and the comprehensiveness of our GAOKAO-Eval benchmark.

## 3.1 DISCREPANCY BETWEEN HIGH SCORES AND HUMAN-ALIGNED CAPABILITIES

We first obtained the score ratings and difficulties, then compared these to the theoretical human performance curves. The results showed significant inconsistencies, as LLMs' difficulty alignment did not match human patterns.

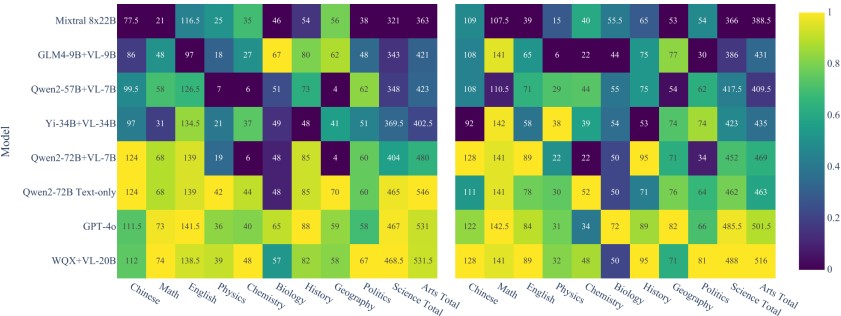

Figure 4: Total performance of LLMs in New Curriculum Standard Paper and National Type A Paper. $+VL$: questions involving images will use the corresponding multimodal version of the model for inference.

**Overall Performance Analysis.** The Figure 4 below present the performance scores of various models on the New Curriculum Standard Paper and the National Type A Paper, ranked by Science Total Score.

**Difficulty of Questions.** Human evaluators design tests to follow a normal distribution in difficulty. To assess LLMs' alignment with this principle, we designed a hybrid approach combining manual annotations with an Elo rating system, which incorporates both human expertise and LLM-based judgments. This system adjusts LLM scores based on pairwise comparisons, allowing us to evaluate question difficulty and model performance consistency. The refined difficulty ratings closely align with human expert judgments, with an internal correlation of up to 0.94 (Figure 5).

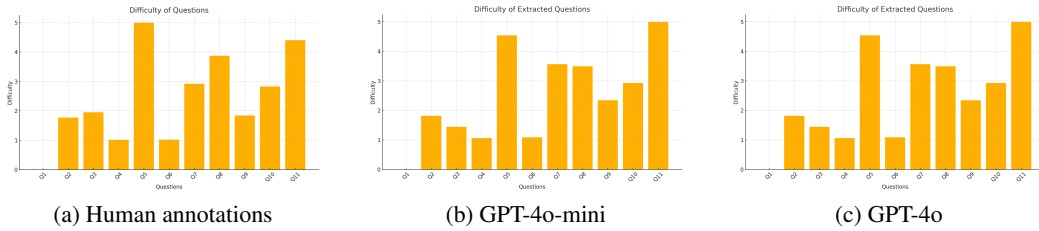

| (a) Human annotations | (b) GPT-4o-mini | (c) GPT-4o |

Figure 5: Consistency distribution of Elo ratings across different models and methods, demonstrating alignment with human expert difficulty ratings.

**Comparison with human performance using Rasch model.** This subsection explores the performance of LLMs in comparison to human performance using the Rasch model, a common method in education and psychometrics for evaluating the relationship between test item difficulty and the probability of a correct response Boone & Noltemeyer (2017); Khine (2020).

The Rasch model (Rasch, 1993), benchmarks its measurements against objective standards to ensure reliability and objectivity, which is particularly useful in assessing whether LLMs can replicate the expected human-aligned response patterns across different difficulty levels. According to the principles of the Rasch model, the probability of a specific individual responding correctly to a specific item can be represented by a function of the individual's ability and the item's difficulty:

$$P(X = 1|\theta, b) = \frac{e^{\theta-b}}{1 + e^{\theta-b}} \quad (1)$$

where $P(X = 1|\theta, b)$ represents the probability that an examinee with ability level $\theta$ will answer an item correctly. $\theta$ represents the examinee's ability level, and $b$ represents the difficulty of the item. In this study, we directly use this equation as the basis for evaluation.

As shown in Figure 6, the relationship between question difficulty and scoring rate predicted by the Rasch model demonstrates a poor fit to the real data, as evidenced by the low R-squared value of -0.23. This low R-squared value suggests *a significant mismatch between the LLMs' capabilities and the expected human-aligned ability*. LLMs struggles to consistently align its performance with the varying difficulty of the questions.

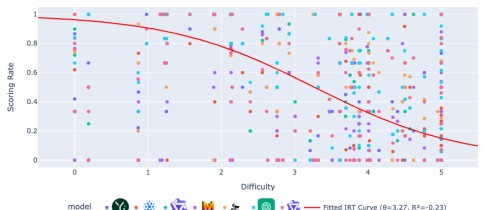

Figure 6: The fitted IRT curve for evaluating LLM performance. The x-axis represents the difficulty level of questions, while the y-axis represents the scoring rate $S$. The red line represents the fitted curve, indicating how well the Rasch model fits the observed data.

### 3.2 LLMs' UNIQUE ERROR PATTERNS

This subsection explores the unique scoring patterns exhibited by LLMs when evaluated across various question difficulties. Understanding these patterns is essential for developing robust evaluation metrics that align with human judgment and accurately reflect LLM capabilities.

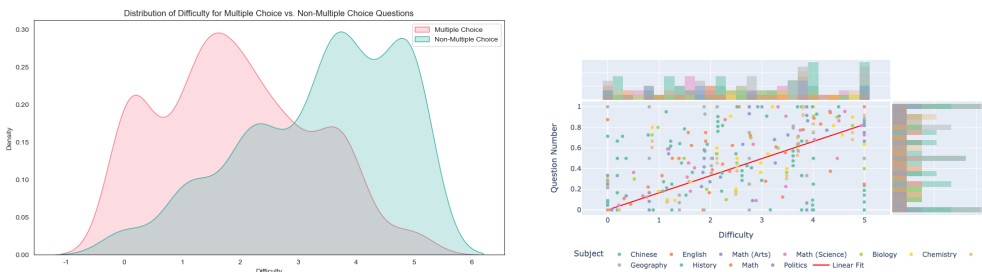

(a) Relationship between question difficulty and question types.

(b) Correlation between question difficulty and question order.

Figure 7: Analysis of question difficulty in relation to question types and order.

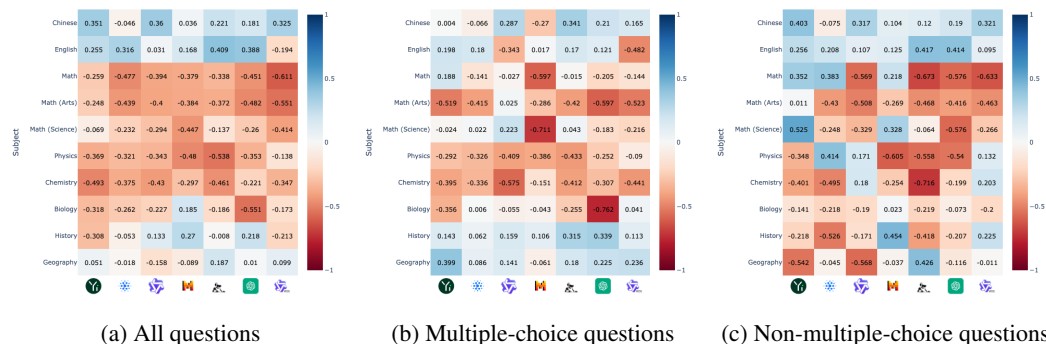

(a) All questions     (b) Multiple-choice questions     (c) Non-multiple-choice questions

Figure 8: Correlation heatmaps between difficulty and scoring rate for all questions, multiple-choice questions, and non-multiple-choice questions.

**Analysis of Difficulty Ratings.** Next, we delve into the characteristics of these difficulty ratings to understand their implications. Figure 7a illustrates the relationship between difficulty and question types, showing that certain question types consistently rank as more difficult across models. Figure 7b explores how difficulty correlates with the order of questions within a test. These analyses indicate that our difficulty ratings are well-aligned with human perception and accurately reflect the human-aligned capabilities of LLMs, further validating the reliability of GAOKAO-Eval's evaluation framework. These findings collectively suggest that our use of the Elo rating system, combined with human judgment, provides a robust and human-aligned approach to assessing LLM performance across varying question difficulties.

**Semi Difficulty-Invariant Scoring Distribution.** We define the phenomenon of *Semi difficulty-invariant scoring distribution* as the lack of significant correlation between question difficulty and the scoring rate of language models. Mathematically, we compute the Pearson correlation coefficient between question difficulties $b_s$ and the observed scoring rates $P_{ij}(s)$ for subject $i$ and model $j$:

$$\rho_{b,P}^{(ij)} = \frac{Cov(b, P_{ij})}{\sigma_b \sigma_{P_{ij}}} \qquad (2)$$

where $Cov(b, P_{ij})$ is the covariance between question difficulty and scoring rate, and $\sigma_b$ and $\sigma_{P_{ij}}$ are the standard deviations of question difficulty and scoring rate, respectively. A small absolute value of $\rho_{b,P}^{(ij)}$ indicates that the scoring rate is approximately invariant with respect to question difficulty.

Figure 8 presents three correlation heatmaps illustrating the relationship between question difficulty and scoring rate across different subjects and language models. The correlations are calculated separately for all questions. The low correlation coefficients observed confirm the semi difficulty-invariant nature of the scoring distributions for the language models studied.

**High Variance in Performance on Similar Difficulty Questions.** Our analysis reveals high variance in LLM scoring rates for questions of similar difficulty, as shown in Figure 6. This phenomenon

deviates from the expected monotonic decrease in performance as difficulty increases predicted by Item Response Theory (IRT). The fitted IRT curves poorly capture the data distribution, indicated by low $R^2$ values (-0.22). This misfit suggests that traditional IRT models, effective for human performance, may not adequately characterize LLM behavior. The high variance limits their reliability in critical applications. To quantify this, we calculate the variance of scoring rates $V_b^{(ij)}$ for subject $i$ and model $j$ within a small difficulty range centered at $b$:

$$V_b^{(ij)} = Var\{P_{ij}(s) \mid b_s \in [b - \Delta b, b + \Delta b]\} \tag{3}$$

### 3.3 LLMs' UNIQUE SCORING PATTERNS

As illustrated in Figure 10, LLMs frequently produce outputs that deviate from human common sense, posing challenges for experts tasked with evaluating these responses. Therefore, the high scores obtained by the model only indicate accuracy according to the specific grading rules used, but do not necessarily reflect a high level of human-like capability. For instance, in Figure 10 a, the model infers a vertical relationship through parallel reasoning, which is illogical. In Figure 10 b, despite errors in the intermediate steps, the model manages to guess the correct answer. In Figure 10 c, the model generates an ancient Chinese poem that never existed in historical records. Lastly, in Figure 10 d, the task explicitly requires a brief summary, yet the model copies the entire passage verbatim. These anomalies highlight the inherent difficulties in relying on LLMs for tasks that require a nuanced understanding of context and common sense. From these examples, it is evident that *even when the LLM captures the "key conceptions", this does not necessarily signify a genuine mastery of the question*.

### 3.4 INCONSISTENCIES IN LLM GRADING

We observed that human examiners encountered too high Inconsistent Score Rate (ISR) in over 32% of cases due to the unique scoring patterns of LLMs. A high ISR indicates that human graders are more likely to disagree when assessing LLM-generated answers compared to typical human responses, which exacerbates the phenomenon of *high variance in performance on similarly difficult questions*. As shown in Figure 9, the ISR varies across subjects. Notably, the humanities subjects, such as Politics and History, exhibit higher inconsistency rates compared to science subjects like Physics and Math. This suggests that LLMs may face more challenges in consistently interpreting and responding to questions in humanities, potentially due to the abstract and context-dependent nature of these subjects. For instance, the ISR for Politics reaches up to 41.18% in some models, indicating that nearly half of the responses deviate significantly from the average, highlighting the difficulties LLMs have with the subjective and often nuanced content typical of humanities. In contrast, the ISR for Physics is much lower, often below 20%, reflecting more consistent performance in subjects that rely on more concrete and structured knowledge.

$$ISR_{ij} = \frac{|s \in S_{ij} : |s - \mu_{ij}| > \sigma_{ij}|}{|S_{ij}|} \tag{4}$$

where $S_{ij}$ represents the set of all scores for subject $i$ and model $j$, $\mu_{ij}$ is the mean score, and $\sigma_{ij}$ is the standard deviation of scores for the same subject-model pair.

Just as Gaokao scores for human candidates have inherent variability, LLM evaluation cannot achieve absolute consistency. Therefore, the scores in GAOKAO-Eval should be interpreted with caution, especially when comparing across different subjects or models. To address potential bias, each question was reviewed by at least three experienced teachers, with the average score taken as the final grade. Significant discrepancies were re-evaluated and adjusted to minimize bias. *However, LLM responses tend to be more misleading for human graders.* As with the RM pattern observed by Qiao et al. (2024), models often produce correct final answers despite flawed intermediate steps, which makes grading more challenging for evaluators who rely on process-based scoring. This can result in greater grading discrepancies, as reflected in the ISR data (Figure 9), particularly in subjects like Politics and Math. These inconsistencies arise from LLMs' unique characteristics, leading to greater divergence in human evaluators' judgments.

## 4 DISCUSSION

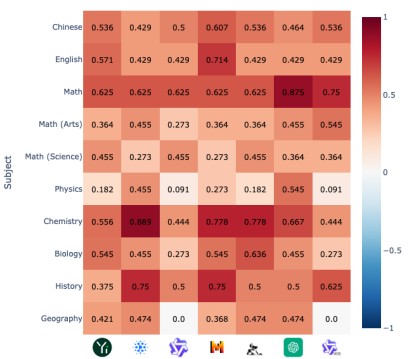

Figure 9: Heatmap of subjects and model performance showing the Inconsistent Score Rate (ISR) across different subjects.

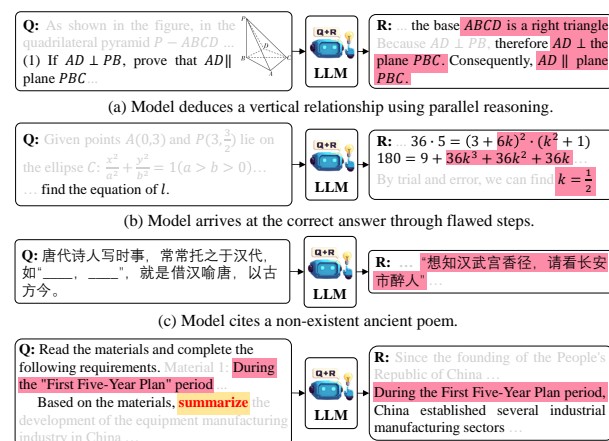

(a) Model deduces a vertical relationship using parallel reasoning.

(b) Model arrives at the correct answer through flawed steps.

(c) Model cites a non-existent ancient poem.

(d) Model fails to summarize as instructed and instead duplicates content.

Figure 10: Unique Error Patterns of the Model.

Our findings through GAOKAO-Eval provide insights into the capabilities and limitations of current LLMs. The observed discrepancy between high benchmark scores and human-aligned cabilities in LLMs aligns with recent studies questioning the reliability of existing evaluation methods (Zellers et al., 2019; Sakaguchi et al., 2020; Lin et al., 2022). Our results extend these concerns to a comprehensive, annually updated benchmark based on real-world educational assessments.

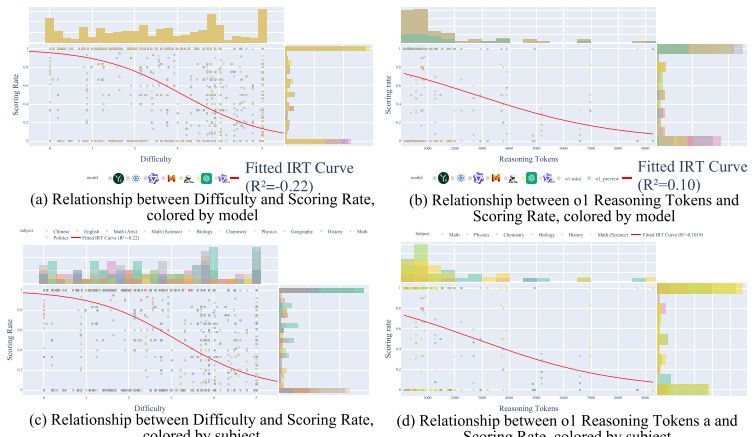

(a) Relationship between Difficulty and Scoring Rate, colored by model

(b) Relationship between o1 Reasoning Tokens and Scoring Rate, colored by model

(c) Relationship between Difficulty and Scoring Rate, colored by subject

(d) Relationship between o1 Reasoning Tokens a and Scoring Rate, colored by subject

Figure 11: Transforming O1's reasoning tokens into human-aligned difficulties improves the fit ($R^2$) with the Rasch model.

**Score and human-aligned capability mismatch in previous benchmarks.** Previous benchmarks may have missed the issue of mismatched scores and human-like capabilities due to their focus on multiple-choice questions with non-continuous score distributions. Additionally, many models likely encountered similar training data (non-leaky), making it difficult to obtain a consistent, continuous scoring range. As a result, earlier benchmarks struggled to highlight this misalignment.

**Towards LLM-Aligned Difficulties.** The semi-invariant scoring behavior of LLMs suggests a critical gap in how these models process information compared to humans. Rather than solving problems through human-like reasoning, LLMs tend to rely on pattern recognition across vast datasets (Chen et al., 2024b), leading to inconsistencies in performance on similarly difficult questions. Figure 11 shows results from external experiment with the latest o1 model. Using o1's reasoning tokens as a proxy for LLM-aligned difficulties improved the fit with the Rasch model. The coefficient of determination ($R^2$) increased from -0.22 to 0.1019. Before we try to improve LLM performance by adopting human-aligned difficulties in benchmarks, it is promising to explore what LLM-aligned difficulties are——challenges that are more suitable for LLMs.

**The High Variance in LLM Performance on the Same Difficulty** raises questions about the consistency and reliability of these models. This variability could have significant implications for deploying LLMs in critical domains where consistent performance is crucial (Zhou et al., 2024).

Figure 11 (b) and (d) suggest that the reduction in variance with more inference tokens indicates a lack of reasoning-related factors contributing to this inconsistency.

## 5 RELATED WORK

**Benchmark for LLMs and VLMs.** Traditional benchmarks have played a pivotal role in assessing the natural understanding capabilities of models (Kahou et al., 2018; Lu et al., 2022a). However, as LLMs evolve, models such as Mistral-Large2 (Mistral AI, 2024), Llama3.1-405B (Dubey et al., 2024), and GPT-4 (OpenAI et al., 2024) begin to outperform human capabilities on these traditional tasks. So there is a growing recognition of the need for more sophisticated benchmarks that can evaluate a broader spectrum of abilities, encompassing both natural language processing (NLP) and multimodal tasks.

MMLU, CMMLU, C-Eval have expanded the evaluation landscape by incorporating a wider range of subjects and difficulty levels, primarily through multiple-choice questions (MCQs) (Hendrycks et al., 2021a; Li et al., 2024; Huang et al., 2023). In addition, Math utilizes a fill-in-the-blank format to measure models' mathematical problem-solving abilities (Hendrycks et al., 2021c). On the multimodal front, benchmarks like MMMU and Math Vista have been introduced to assess models' abilities to integrate and interpret information across different modalities with reasoning, such as text and images(Yue et al., 2024; Lu et al., 2024; Chen et al., 2024a).

Despite these advancements, current benchmarks face notable limitations. The reliance on MCQs, while valuable for scalability and objectivity, often limits the types of questions and scenarios that can be evaluated, potentially overlooking critical aspects of language comprehension and generation. Moreover, both NLP and multimodal benchmarks struggle with the issue of question leakage. Our work introduces a novel benchmark by leveraging the rich and diverse question types and a broad array of subjects and question formats from the GAOKAO examinations.

**Gaokao Evaluation.** Gaokao is a comprehensive national academic test that serves as the primary criterion for university admission in China. Two notable works in this area are Gaokao-bench and GAOKAO-MM (Zong & Qiu, 2024; Zhang et al., 2023). Gaokao-bench compiled a dataset of multiple-choice and subjective questions from previous Gaokao examinations, while GAOKAO-MM expanded this approach to include multimodal elements. Both benchmarks aim to provide a comprehensive evaluation of LLMs' language understanding, reasoning abilities, and multimodal integration capabilities.

The underlying assumption for these two benchmarks is that if LLMs can perform well on tasks that challenge human intelligence, they may be developing more human-like cognitive capabilities. However, our research demonstrates that high scores on these benchmarks do not necessarily equate to human-like abilities in LLMs.

**Error Patterns and Human Evaluation of LLMs.** Berglund et al. (2024) finds a reversal curse pattern, while Wu et al. (2023) designs 11 counterfactual evaluation tasks and observes consistent and substantial degradation of LM performance. Some researchers find that LLMs make a significant number of basic errors in code writing and are unable to complete code with potential bugs (Zhong & Wang, 2023; Jesse et al., 2023; Dinh et al., 2023). Chen et al. (2024b) builds an automated evaluation method and discoverys 8 error patterns. Rather than summarizing error patterns in GAOKAO, our analysis delves deeper into how these patterns contribute to the phenomenon where high scores cannot reflect high capabilities. This suggests a potential gap in reasoning abilities.

## 6 CONCLUSION

GAOKAO-Eval provides a comprehensive and annually updated benchmark for evaluating LLM capabilities. Our study reveals that high scores on existing benchmarks do not necessarily reflect human-aligned reasoning abilities in LLMs. We identifies unique scoring patterns in LLMs, including semi difficulty-invariant distributions and high performance variance on similar difficulty questions. These findings challenge the effectiveness of current evaluation methods and highlight the need for more LLM-aligned difficulties analysis. Future work could focus on developing reasoning-based metrics to better align difficulty assessments with LLM capabilities, addressing the problem of high scores failing to capture true capability.

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

APPENDIX

We provide more details of the proposed method and additional experimental results to help better understand our paper. In summary, this appendix includes the following contents:

APPENDIX CONTENTS

## A    DETAIL OF WQX TRAINING

In this section, we delve into the details surrounding the WQX model, which was specifically developed to validate the comprehensiveness of the Gaokao evaluation system. To achieve this, we embarked on an extensive data preparation process, continuing the pre-training on the foundation provided by the InternLM2-20b-base model. The subsequent paragraphs will outline the datasets utilized by the WQX model, our data preparation methodologies, and the intricate details of the training process.

### A.1   GAOKAO TRAINING DATASET

To construct a comprehensive Gaokao dataset, we embarked on an extensive data collection process, gathering over 25 million examination questions from online resources and educational books. After a meticulous deduplication process, we narrowed down the dataset to 17.5 million unique questions. In addition to these questions, we incorporated more than 1GB of textual material from academic books into our training data to further enrich the dataset's diversity and depth. To enhance the quality and relevance of our dataset for training LLMs specifically for Gaokao preparation, we employed several innovative techniques for data augmentation and retrieval.

**Numerical Reasoning Enhancement**   Given that solution processes for numerical calculation problems often contain omissions, we employed the InternLM2-20b model to supplement and elaborate on these mathematical problem-solving steps. To further bolster the model's numerical computation capabilities, we developed a custom calculation tool similar to SymPy, which provides the correct and detailed step-by-step solutions. This tool was used to synthesize a large volume of computational data, thereby enhancing the model's arithmetic proficiency.

**Chain-of-Thought Augmentation**   For questions lacking detailed explanations, we implemented a multi-step approach using the InternLM2-20b model. In the case of multiple-choice questions, we introduced variations in the model's response order to ensure robustness. For questions that the model consistently answered correctly, we prompted it to generate comprehensive explanations, thus enriching the dataset with detailed problem-solving rationales.

**Enhanced Retrieval and Filtering from Common Crawl**   We employed the Query of CC technique (Fei et al., 2024) to retrieve Gaokao-relevant data from Common Crawl, using examination questions as queries. To ensure data quality, we implemented a novel filtering process: retrieved

data was used as context in a Retrieval-Augmented Generation (RAG) framework. The model reattempted questions using this context, and we retained only the retrieved documents that enabled correct answers to previously mishandled questions. This filtered dataset served as high-quality supplementary knowledge for training, effectively enhancing the model's performance on Gaokao-style examinations.

## A.2 GAOKAO TRAINING PROCESS

WQX model, built upon InternLM2-20b-base, was trained on the curated Gaokao dataset for two complete epochs. We employed the InternEvo framework (Team, 2023), maintaining a context length of 4096 tokens. For content exceeding this length, we split and truncated it into multiple data points. The training configuration utilized mixed-precision computation with bfloat16 and FlashAttention2 for optimal efficiency (Dao, 2024). We used the AdamW optimizer ($\beta_1 = 0.9$, $\beta_2 = 0.95$, weight decay $= 0.1$) with a cosine decay learning rate schedule, peaking at $3 \times 10^{-5}$ after 2000 warm-up steps and decreasing to $3 \times 10^{-6}$. Each training batch consisted of approximately 0.5M tokens, balancing computational efficiency with effective learning.

## B SUPPLEMENTARY EXPLANATION OF GAOKAO-EVAL

### B.1 MORE DETAIL ABOUT GAOKAO

**Paper Type Diversity**   The comprehensive nature of GAOKAO-Eval is further enhanced by its coverage of multiple paper types, reflecting the diverse educational landscape across China. As detailed in Table 1, the benchmark includes various Gaokao models, such as the "3+1+2" New Pattern, the "3+3" Pattern, and the Traditional Pattern. With the reform of the Gaokao (National College Entrance Examination) in 2024, there are six types of Gaokao papers nationwide. The Beijing, Shanghai, and Tianjin papers, along with the National paper A, cover all subjects. Provinces using the New Curriculum Standard I and II papers use corresponding Chinese, Mathematics, and English papers, while most provinces independently set their own exams for other subjects. In GAOKAO-Eval, we tested all publicly available papers from the New Curriculum and National paper A. Regarding the Gaokao models, the current system is primarily divided into three major categories:

- **The "3+1+2" New Pattern**   widely adopted by 23 provinces, is structured around the core subjects of Chinese, Mathematics, and English. Students are required to choose either Physics or History as their primary subject and select two additional subjects from the remaining four (Political Science, Geography, Chemistry, Biology).

- **The "3+3" Pattern**   currently used by 6 provinces, allows students, after completing the core subjects (Chinese, Mathematics, and English), to freely choose three elective subjects from six options (Political Science, Geography, Chemistry, Biology, and in Zhejiang, an additional subject of Technology).

- **The Traditional Pattern**   The remaining 5 provinces still adhere to the traditional subject division system of the **National Paper A**, maintaining the conventional academic assessment pathway.

**Distribution of Questions in GAOKAO**   As detailed in Table 2, the distribution of question types across various subjects showcases the comprehensive scope of the GAOKAO-Eval benchmark.

### B.2 EXAMPLES OF QUESTIONS AND EXPLANATIONS

In this section, we present some examples of questions along with their answers and explanations. As shown in Figure 1, there are questions about the spatial distribution characteristics of traditional dwellings, their designs, and the roles of public spaces in Shuangfeng Village. Additionally, Figure 2 demonstrates the predictions made by GPT-4o on these questions.

| Paper Type | Provinces/Cities Using It |
|---|---|
| New Curriculum Standard Paper I | Guangdong, Fujian, Hubei, Hunan, Jiangsu, Hebei, Shandong, Zhejiang, Jiangxi, Anhui, Henan |
| New Curriculum Standard Paper II | Liaoning, Chongqing, Hainan, Shanxi, Xinjiang, Guangxi, Guizhou, Heilongjiang, Gansu, Jilin, Yunnan, Tibet |
| New Curriculum | Shanxi, Henan, Yunnan, Tibet, Xinjiang |
| National Paper A | Sichuan, Inner Mongolia, Ningxia, Shaanxi, Qinghai |
| Beijing Paper | Beijing |
| Shanghai Paper | Shanghai |
| Tianjin Paper | Tianjin |

Table 1: Gaokao Paper Types and Their Usage.

Table 2: Distribution of Question Types Across Subjects in GAOKAO-Eval 2024.

| Subject | Question Type | Count | Percentage (%) |
|---|---|---|---|
| Chemistry | Multiple Choice | 14 | 60.87 |
| | Fill in the Blanks | 7 | 30.43 |
| | Optional Fill in the Blanks | 2 | 8.70 |
| History | Multiple Choice | 24 | 75.00 |
| | Short Answer | 8 | 25.00 |
| Geography | Multiple Choice | 22 | 53.66 |
| | Short Answer | 17 | 41.46 |
| | Optional Short Answer | 2 | 4.88 |
| Politics | Multiple Choice | 12 | 71.79 |
| | Short Answer | 5 | 28.21 |
| Mathematics | Multiple Choice | 8 | 42.11 |
| | Fill in the Blanks | 3 | 15.79 |
| | Multiple Selection | 3 | 15.79 |
| | Short Answer | 5 | 26.32 |
| Mathematics (Arts) | Multiple Choice | 12 | 52.17 |
| | Fill in the Blanks | 4 | 17.39 |
| | Short Answer | 7 | 30.43 |
| Mathematics (Science) | Multiple Choice | 12 | 52.17 |
| | Fill in the Blanks | 4 | 17.39 |
| | Short Answer | 7 | 30.43 |
| Physics | Multiple Choice | 13 | 44.83 |
| | Fill in the Blanks | 4 | 13.79 |
| | Multiple Selection | 3 | 10.34 |
| | Short Answer | 5 | 17.24 |
| | Optional Questions | 2 | 6.90 |
| | Optional Multiple Choice | 2 | 6.90 |
| Biology | Multiple Choice | 12 | 52.17 |
| | Fill in the Blanks | 9 | 39.13 |
| | Optional Fill in the Blanks | 2 | 8.70 |
| English | 7 out of 5 | 2 | 11.11 |
| | Writing - Composition | 1 | 5.56 |
| | Writing - Error Correction | 1 | 5.56 |
| | Long Composition | 1 | 5.56 |
| | Cloze Test | 2 | 11.11 |
| | Short Composition | 1 | 5.56 |
| | Grammar | 1 | 5.56 |
| | Grammar Completion | 1 | 5.56 |
| | Reading Comprehension | 8 | 44.46 |
| Chinese | Composition | 2 | 4.44 |
| | Classical Poetry Reading | 4 | 8.89 |
| | Famous Passage Recitation | 2 | 4.44 |
| | Classical Chinese Reading | 9 | 20.00 |
| | Modern Text Reading | 18 | 40.00 |
| | Language Usage | 10 | 22.22 |

## Examples of Questions.

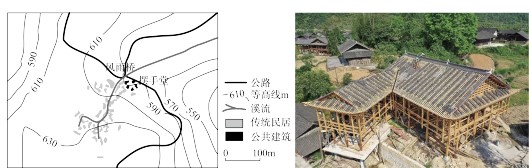

**Question**: Shuangfeng Village in Yongshun County, Hunan Province, is a typical Tujia village located in the hinterland of the Wuling Mountains. A small stream meanders through the village, accompanied by the sound of flowing water, winding stone paths, and Tujia corner-stilt houses built along the mountain, creating a beautiful picture of the Tujia mountain village, known as the "First Tujia Village in China." The left figure shows the distribution of buildings in Shuangfeng Village, and the right figure shows the corner-stilt house under construction. Based on this, answer the following questions.

1. The spatial distribution characteristics of traditional dwellings in Shuangfeng Village are ( )
   A. Along the contour lines
   B. Along the stream
   C. Around public buildings
   D. Along the road

2. The corner design of traditional dwellings in Shuangfeng Village is mainly to ( )
   A. Adapt to the humid environment
   B. Facilitate local materials
   C. Make full use of space
   D. Pursue visual aesthetics

3. The Baishoutang (including the Tuwang Temple) and the Wind and Rain Bridge provide places for public activities and communication for the villagers of Shuangfeng Village, reflecting the public space of the village
   ① Balance ② Economy ③ Public welfare ④ Culture

   A. ①②
   B. ①③
   C. ②④
   D. ③④

**Correct Answer:** B C D

Figure 1: Examples of MM Geo Questions and Answers.

**Examples of Model Prediction.**

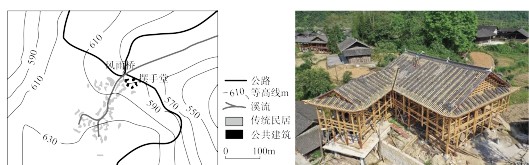

**GPT Prediction:**

- From the left figure, it can be seen that the traditional dwellings in Shuangfeng Village are mostly distributed along the stream, B is correct; from the left figure, it can be seen that the traditional dwellings cross three contour lines and are not parallel to the contour lines, A is incorrect; public buildings are distributed downstream of the river, with only a few scattered households nearby, C is incorrect; the road was built in the later stage of social and economic development, and the traditional dwellings were built earlier, so they could not be distributed along the road, D is incorrect. Hence, B is correct.

- From the figure, it can be seen that Yongshun County in Hunan Province is located in a mountainous area with limited flat land and building space. The corner design of traditional dwellings in Shuangfeng Village is mainly to improve land use efficiency and make full use of space, C is correct; adopting the stilt design provides good ventilation and adapts to the humid geographical environment, A is incorrect; the main structure of the buildings uses bamboo and wood, which are abundant locally, facilitating local materials, B is incorrect; the initial design of traditional dwellings was to adapt to the local natural geographical environment, not to pursue visual aesthetics, D is incorrect. Hence, C is correct.

- The Baishoutang (including the Tuwang Temple) and the Wind and Rain Bridge provide places for public activities and communication for the villagers of Shuangfeng Village, reflecting the public welfare and cultural nature of the village's public space, ③④ are correct; the Wind and Rain Bridge and Baishoutang are mainly places for villagers' activities and are not focused on economic activities, so they do not reflect the economic nature of the village's public space, ② is incorrect; the Baishoutang and Wind and Rain Bridge are concentrated downstream of the stream, so they do not reflect balance, ① is incorrect. Hence, D is correct, and ABC are incorrect. Therefore, D is correct.

Figure 2: Examples of GPT-4o Predictions.

## B.3 OVERALL PERFORMANCE ANALYSIS

The tables below present the performance scores of various models on the New Curriculum Standard Paper and the National Paper A, ranked by Science Total Score.

Table 3: New Curriculum Standard Paper [†] Scores (Ranked by Science Total Score)

| Model | Chinese | Math | English | Physics | Chemistry | Biology | History | Geography | Politics | Science Total | Arts Total |
|---|---|---|---|---|---|---|---|---|---|---|---|
| WQX-20B+VL-20B | 112 | **74** | 138.5 | 39 | **48** | 57 | 82 | 58 | **67** | **468.5** | 531.5 |
| GPT-4o | 111.5 | 73 | **141.5** | 36 | 40 | **65** | **88** | 59 | 58 | 467 | 531 |
| Qwen2-72B Text-only | **124** | 68 | 139 | **42** | 44 | 48 | 85 | **70** | 60 | 465 | **546** |
| Qwen2-72B+VL-7B | **124** | 68 | 139 | 19 | 6 | 48 | 85 | 4 | 60 | 404 | 480 |
| Yi-34B+VL-34B | 97 | 31 | 134.5 | 21 | 37 | 49 | 48 | 41 | 51 | 369.5 | 402.5 |
| Qwen2-57B+VL-7B | 99.5 | 58 | 126.5 | 7 | 6 | 51 | 73 | 4 | 62 | 348 | 423 |
| GLM4-9B+VL-9B | 86 | 48 | 97 | 18 | 27 | 67 | 80 | 62 | 48 | 343 | 421 |
| Mixtral 8x22B | 77.5 | 21 | 116.5 | 25 | 35 | 46 | 54 | 56 | 38 | 321 | 363 |

$+VL$ means that questions involving images will use the corresponding multimodal version of the model for inference; if there is no "+VL", only pure text inference without images is performed.

Table 4: National Paper A Scores (Ranked by Science Total Score)

| Model | Chinese | English | Math (Science) | Physics | Chemistry | Biology | Math (Arts) | History | Geography | Science Total | Arts Total (Excl. Politics) |
|---|---|---|---|---|---|---|---|---|---|---|---|
| Qwen2-72B Text-only | **128** | 141 | **89** | 32 | 48 | 50 | 95 | 71 | **81** | **488** | **516** |
| GPT-4o | 122 | **142.5** | 84 | 31 | 34 | **72** | **89** | **82** | 66 | 485.5 | 501.5 |
| WQX-20B+VL-20B | 111 | 141 | 78 | 30 | **52** | 50 | 71 | 76 | 64 | 462 | 463 |
| Qwen2-72B+VL-7B | **128** | 141 | 89 | 22 | 22 | 50 | 95 | 71 | 34 | 452 | 469 |
| Mixtral 8x22B | 92 | 142 | 58 | **38** | 39 | 54 | 53 | 74 | 74 | 423 | 435 |
| GLM4-9B+VL-9B | 108 | 110.5 | 71 | 29 | 44 | 55 | 75 | 54 | 62 | 417.5 | 409.5 |
| Qwen2-57B+VL-7B | 108 | 141 | 65 | 6 | 22 | 44 | 75 | 77 | 30 | 386 | 431 |
| Yi-34B+VL-34B | 109 | 107.5 | 39 | 15 | 40 | 55.5 | 65 | 53 | 54 | 366 | 388.5 |

$+VL$ means that questions involving images will use the corresponding multimodal version of the model for inference; if there is no "+VL", only pure text inference without images is performed.

### B.4 DETAIL OF SUBJECT SCORES

This section meticulously details the performance scores of models across a broad spectrum of subjects, showcasing their capabilities and limitations within an academic context. These assessments span across both the New Curriculum and the National Paper A, covering a wide range of disciplines including Chinese, Mathematics (distinguished between Science and Arts tracks), English, Physics, Chemistry, Biology, History, Geography, and Politics.

### B.4.1 NEW CURRICULUM

**Chinese**

Table 5: Scores for Different Sections in Chinese.

| Model | Modern Reading (35) | Classical Reading (22) | Poetry Reading (9) | Quote Writing (6) | Language Use (18) | Essay (60) | Total (150) |
|---|---|---|---|---|---|---|---|
| Qwen2-72B | 31 | 19 | 9 | 6 | 9 | 50 | 124 |
| WQX-20B | 30 | 17 | 6 | 6 | 7 | 46 | 112 |
| GPT-4o | 32 | 10 | 8 | 2 | 9 | 50.5 | 111.5 |
| Qwen2-57B | 27 | 12 | 7 | 6 | 2 | 45.5 | 99.5 |
| Yi-1.5-34B | 28 | 8 | 5 | 2 | 4 | 50 | 97 |
| GLM4-9B | 21 | 6 | 8 | 6 | 4 | 41 | 86 |
| Mixtral 8x22B | 18 | 3 | 7 | 2 | 3 | 44.5 | 77.5 |

**Mathematics**

Table 6: Score Distribution for Each Question Type in Mathematics

| Model | Single Choice Questions (40) | Multiple Choice Questions (18) | Fill-in-the-Blank Questions (15) | Short Answer Questions (77) | Total Score (150) |
|---|---|---|---|---|---|
| WQX-20B | 30 | 8 | 10 | 26 | 74 |
| GPT-4o | 35 | 6 | 10 | 22 | 73 |
| Qwen2-72B | 30 | 10 | 10 | 18 | 68 |
| Qwen2 57B | 35 | 9 | 5 | 9 | 58 |
| GLM4-9B | 30 | 6 | 0 | 12 | 48 |
| Yi-1.5-34B | 20 | 7 | 0 | 4 | 31 |
| Mixtral 8x22B | 10 | 0 | 0 | 11 | 21 |

**English**

Table 7: Score Distribution for Each Question Type in English

| Model | Listening (30) | Reading Comprehension (37.5) | Choose 5 out of 7 (12.5) | Cloze Test (15) | Grammar Completion (15) | Writing (40) | Total Score (150) |
|---|---|---|---|---|---|---|---|
| GPT-4o | 30 | 37.5 | 10 | 14 | 15 | 35 | 141.5 |
| Qwen2-72B | 30 | 35 | 12.5 | 14 | 13.5 | 34 | 139 |
| WQX-20B | 30 | 37.5 | 10 | 15 | 13.5 | 32.5 | 138.5 |
| Yi-1.5-34B | 30 | 35 | 10 | 11 | 13.5 | 35 | 134.5 |
| Qwen2 57B | 30 | 35 | 10 | 9 | 15 | 27.5 | 126.5 |
| Mixtral 8x22B | 30 | 37.5 | 5 | 2 | 9 | 33 | 116.5 |
| GLM4-9B | 30 | 35 | 0 | 6 | 6 | 20 | 97 |

## Physics

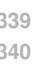

Table 8: Score Distribution for Each Question Type in Physics

| Model | Single Choice Questions (30) | Multiple Choice Questions (18) | Fill-in-the-Blank Questions (18) | Short Answer Questions (44) | Total Score (110) |
|---|---|---|---|---|---|
| Qwen2-72B | 18 | 6 | 6 | 12 | 42 |
| WQX-20B+VL-20B | 18 | 12 | 9 | 0 | 39 |
| GPT-4o | 18 | 6 | 2 | 10 | 36 |
| Mixtral 8x22B | 12 | 3 | 5 | 5 | 25 |
| Yi-1.5-34B+VL-34B | 12 | 6 | 2 | 1 | 21 |
| Qwen2-72B+VL-7B | 18 | 0 | 0 | 1 | 19 |
| GLM4-9B+4v-9B | 12 | 3 | 2 | 1 | 18 |
| Qwen2-57B+VL-7B | 6 | 0 | 0 | 1 | 7 |

## Chemistry

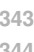

Table 9: Score Distribution for Each Question Type in Chemistry

| Model | Multiple Choice Questions (42) | Fill-in-the-Blank Questions (58) | Total Score (100) |
|---|---|---|---|
| WQX-20B+VL-20B | 18 | 30 | 48 |
| Qwen2-72B | 18 | 26 | 44 |
| GPT-4o | 18 | 22 | 40 |
| Yi-1.5-34B+VL-34B | 24 | 13 | 37 |
| Mixtral 8x22B | 18 | 17 | 35 |
| GLM4-9B+4v-9B | 12 | 15 | 27 |
| Qwen2-72B+VL-7B | 6 | 0 | 6 |
| Qwen2-57B+VL-7B | 6 | 0 | 6 |

## Biology

Table 10: Score Distribution for Each Question Type in Biology

| Model | Multiple Choice Questions (36) | Fill-in-the-Blank Questions (54) | Total Score (90) |
|---|---|---|---|
| GLM4-9B+4v-9B | 36 | 31 | 67 |
| GPT-4o | 30 | 35 | 65 |
| WQX-20B+VL-20B | 24 | 33 | 57 |
| Qwen2-57B+VL-7B | 30 | 21 | 51 |
| Yi-1.5-34B+VL-34B | 18 | 31 | 49 |
| Qwen2-72B+VL-7B | 30 | 18 | 48 |
| Mixtral 8x22B | 6 | 40 | 46 |

## History

Table 11: Score Distribution for Each Question Type in History

| Model | Multiple Choice Questions (48) | Short Answer Questions (52) | Total Score (100) |
|---|---|---|---|
| GPT-4o | 44 | 44 | 88 |
| Qwen2-72B+VL-7B | 40 | 45 | 85 |
| WQX-20B+VL-20B | 44 | 38 | 82 |
| GLM4-9B+4v-9B | 48 | 32 | 80 |
| Qwen2-57B+VL-7B | 44 | 29 | 73 |
| Mixtral 8x22B | 12 | 42 | 54 |
| Yi-1.5-34B+VL-34B | 48 | 0 | 48 |

**Geography**

Table 12: Score Distribution for Each Question Type in Geography.

| Model | Multiple Choice Questions (44) | Short Answer Questions (56) | Total Score (100) |
|---|---|---|---|
| Qwen2-72B | 32 | 38 | 70 |
| GLM4-9B+4v-9B | 36 | 26 | 62 |
| GPT-4o | 32 | 27 | 59 |
| WQX-20B+VL-20B | 32 | 26 | 58 |
| Mixtral 8x22B | 24 | 32 | 56 |
| Yi-1.5-34B+VL-34B | 20 | 21 | 41 |
| Qwen2-72B+VL-7B | 4 | 0 | 4 |
| Qwen2-57B+VL-7B | 4 | 0 | 4 |

**Politics**

Table 13: Score Distribution for Each Question Type in Politics.

| Model | Multiple Choice Questions (48) | Short Answer Questions (52) | Total Score (100) |
|---|---|---|---|
| WQX-20B+VL-20B | 40 | 27 | 67 |
| Qwen2-57B+VL-7B | 36 | 26 | 62 |
| Qwen2-72B+VL-7B | 40 | 20 | 60 |
| GPT-4o | 44 | 14 | 58 |
| Yi-1.5-34B+VL-34B | 36 | 15 | 51 |
| GLM4-9B+4v-9B | 36 | 12 | 48 |
| Mixtral 8x22B | 28 | 10 | 38 |

### B.4.2 NATIONAL PAPER A

**Chinese**

Table 14: Score Distribution for Each Question Type in Chinese.

| Model | Modern Text Reading (36) | Classical Chinese Reading (19) | Ancient Poetry Reading (9) | Memorization of Famous Works and Quotations (6) | Language and Text Application (20) | Essay (60) | Total Score (150) |
|---|---|---|---|---|---|---|---|
| Qwen2-72B | 35 | 19 | 9 | 2 | 15 | 48 | 128 |
| GPT-4o | 29 | 19 | 8 | 4 | 14 | 48 | 122 |
| WQX-20B | 26 | 14 | 7 | 6 | 15 | 43 | 111 |
| Yi-1.5-34B | 28 | 12 | 7 | 0 | 16 | 46 | 109 |
| GLM4-9B | 24 | 13 | 8 | 2 | 15 | 46 | 108 |
| Qwen2-57B | 27 | 14 | 7 | 2 | 14 | 44 | 108 |
| Mixtral 8x22B | 24 | 0 | 7 | 0 | 14 | 47 | 92 |

**Mathematics (Science)**

Table 15: Score Distribution for Each Question Type in Mathematics (Science).

| Model | Single Choice Questions (60) | Fill-in-the-Blank Questions (20) | Short Answer Questions (60) | Elective Questions - Short Answer Questions (20) | Total Score (150) |
|---|---|---|---|---|---|
| Qwen2-72B | 50 | 10 | 19 | 15 | 89 |
| GPT-4o | 35 | 15 | 27 | 12 | 84 |
| WQX-20B | 35 | 5 | 38 | 0 | 78 |
| GLM4-9B | 35 | 5 | 28 | 3 | 71 |
| Qwen2-57B | 40 | 5 | 13 | 13 | 65 |
| Mixtral 8x22B | 30 | 0 | 21 | 12 | 58 |
| Yi-1.5-34B | 20 | 0 | 17 | 2 | 39 |

**Mathematics (Arts)**

Table 16: Score Distribution for Each Question Type in Mathematics (Arts)

| Model | Single Choice Questions (60) | Fill-in-the-Blank Questions (20) | Short Answer Questions (60) | Elective Questions - Short Answer Questions (20) | Total Score (150) |
|---|---|---|---|---|---|
| Qwen2-72B | 50 | 15 | 20 | 14 | 95 |
| GPT-4o | 40 | 15 | 24 | 10 | 89 |
| GLM4-9B | 35 | 10 | 27 | 3 | 75 |
| Qwen2-57B | 40 | 10 | 18 | 9 | 75 |
| WQX-20B | 30 | 15 | 26 | 0 | 71 |
| Yi-1.5-34B | 25 | 5 | 31 | 6 | 65 |
| Mixtral 8x22B | 30 | 5 | 15 | 3 | 53 |

**English**

Table 17: Score Distribution for Each Question Type in English.

| Model | Reading Comprehension (30) | Choose 5 out of 7 (10) | Cloze Test (30) | Grammar Completion (15) | Writing (35) | Listening (30) | Total Score (150) |
|---|---|---|---|---|---|---|---|
| GPT-4o | 30 | 10 | 28.5 | 15 | 29 | 30 | 142.5 |
| Mixtral 8x22B | 30 | 10 | 30 | 15 | 27 | 30 | 142 |
| Qwen2-72B | 30 | 10 | 30 | 15 | 26 | 30 | 141 |
| WQX-20B | 30 | 10 | 28.5 | 15 | 27.5 | 30 | 141 |
| Qwen2-57B | 28 | 10 | 30 | 15 | 28 | 30 | 141 |
| GLM4-9B | 26 | 0 | 21 | 12 | 21.5 | 30 | 110.5 |
| Yi-1.5-34B | 24 | 8 | 16.5 | 13.5 | 15.5 | 30 | 107.5 |

**Physics**

Table 18: Score Distribution for Each Question Type in Physics.

| Model | Multiple Choice Questions (48) | Fill-in-the-Blank Questions (15) | Short Answer Questions (32) | Elective Questions - Multiple Choice (10) | Elective Questions (20) | Total Score (110) |
|---|---|---|---|---|---|---|
| Mixtral 8x22B | 27 | 1 | 9 | 1 | 0 | 38 |
| Qwen2-72B | 18 | 1 | 9 | 0 | 4 | 32 |
| GPT-4o | 15 | 5 | 10 | 1 | 0 | 31 |
| WQX-20B+VL-20B | 24 | 1 | 4 | 1 | 0 | 30 |
| GLM4-9B+4v-9B | 18 | 2 | 6 | 2 | 1 | 29 |
| Qwen2-72B+VL-7B | 12 | 2 | 8 | 0 | 0 | 22 |
| Yi-1.5-34B+VL-34B | 9 | 0 | 6 | 0 | 0 | 15 |
| Qwen2-57B+VL-7B | 0 | 2 | 4 | 0 | 0 | 6 |

**Chemistry**

Table 19: Score Distribution for Each Question Type in Chemistry.

| Model | Multiple Choice Questions (42) | Fill-in-the-Blank Questions (43) | Elective Questions - Fill-in-the-Blank Questions (30) | Total Score (100) |
|---|---|---|---|---|
| WQX-20B+VL-20B | 30 | 15 | 10 | 52 |
| Qwen2-72B | 24 | 13 | 13 | 48 |
| GLM4-9B+4v-9B | 24 | 15 | 7 | 44 |
| Yi-1.5-34B+VL-34B | 24 | 13 | 4 | 40 |
| Mixtral 8x22B | 24 | 8 | 7 | 39 |
| GPT-4o | 12 | 14 | 8 | 34 |
| Qwen2-72B+VL-7B | 12 | 7 | 5 | 22 |
| Qwen2-57B+VL-7B | 12 | 7 | 5 | 22 |

**Biology**

Table 20: Score Distribution for Each Question Type in Biology.

| Model | Multiple Choice Questions (36) | Fill-in-the-Blank Questions (39) | Elective Questions - Fill-in-the-Blank Questions (30) | Total Score (90) |
|---|---|---|---|---|
| GPT-4o | 30 | 27 | 23 | 72 |
| Yi-1.5-34B+VL-34B | 30 | 10.5 | 26 | 55.5 |
| GLM4-9B+4v-9B | 24 | 16 | 19 | 55 |
| Mixtral 8x22B | 18 | 21 | 24 | 54 |
| Qwen2-72B+VL-7B | 18 | 17 | 15 | 50 |
| WQX-20B+VL-20B | 18 | 21 | 21 | 50 |
| Qwen2-57B+VL-7B | 18 | 11 | 15 | 44 |

**History**

Table 21: Score Distribution for Each Question Type in History.

| Model | Multiple Choice Questions (48) | Short Answer Questions (52) | Total Score (100) |
|---|---|---|---|
| GPT-4o | 36 | 46 | 82 |
| Qwen2-57B+VL-7B | 40 | 37 | 77 |
| WQX-20B+VL-20B | 40 | 36 | 76 |
| Mixtral 8x22B | 36 | 38 | 74 |
| Qwen2-72B+VL-7B | 32 | 39 | 71 |
| GLM4-9B+4v-9B | 20 | 34 | 54 |
| Yi-1.5-34B+VL-34B | 20 | 33 | 53 |

**Geography**

Table 22: Score Distribution for Each Question Type in Geography.

| Model | Multiple Choice Questions (44) | Short Answer Questions (46) | Elective Questions - Short Answer (10) | Total Score (100) |
|---|---|---|---|---|
| Qwen2-72B | 40 | 31 | 10 | 81 |
| Mixtral 8x22B | 36 | 30 | 8 | 74 |
| GPT-4o | 32 | 24 | 10 | 66 |
| WQX-20B+VL-20B | 24 | 36 | 4 | 64 |
| GLM4-9B+4v-9B | 24 | 28 | 10 | 62 |
| Yi-1.5-34B+VL-34B | 28 | 16 | 10 | 54 |
| Qwen2-72B+VL-7B | 24 | 0 | 10 | 34 |
| Qwen2-57B+VL-7B | 16 | 0 | 14 | 30 |

## B.5 MODEL DETAIL

We restricted our selection of open-source models to those released before June 6, 2024, and included GPT-4o as a benchmark, currently the most powerful model available. Here is an overview of the participating models:

| Name | Developer | Type | Description | Weights Upload Date |
|---|---|---|---|---|
| Qwen2-72B | Alibaba | Language Model | The largest dialogue model in Alibaba's Qwen2 series.[1] | 2024.05.28 |
| Qwen2-57B | Alibaba | Language Model | A MoE dialogue model in Alibaba's Qwen2 series.[2] | 2024.05.04 |
| QwenVL-7B | Alibaba | Multimodal Model | A multimodal dialogue model by Alibaba.[3] | 2023.09.25 |
| Yi-1.5-34B | 01.AI | Language Model | The largest model in the Yi 1.5 series by Wuhan ZeroOne.[4] | 2024.05.12 |
| Yi-VL-34B | 01.AI | Multimodal Model | A multimodal large model by Wuhan 01.AI.[5] | 2024.01.19 |
| GLM4-9B | ZHIPU AI | Language Model | An open-source version of the latest pre-trained model in ZHIPU AI's GLM-4 series.[6] | 2024.06.04 |
| GLM-4v-9B | ZHIPU AI | Multimodal Model | A multimodal model in ZHIPU AI's GLM-4 series.[7] | 2024.06.04 |
| Mixtral 8x22B | Mistral (France) | Language Model | The most powerful dialogue model currently open-sourced by the French AI startup Mistral.[8] | 2024.04.17 |
| GPT-4o | OpenAI (USA) | Multimodal Model | The most powerful model released by OpenAI, currently the world's leading large model.[9] | 2024.05.13 |

Table 23: Summary of Models Participating in Evaluation

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
