# OpenReview forum: "GAOKAO-Eval: Does High Scores Truly Reflect Strong Capabilities in LLMs?"
_ICLR.cc/2025/Conference — Submitted to ICLR 2025_

### Official Review · Reviewer_g5fT · 2024-10-16

**Soundness:** 3
**Presentation:** 2
**Contribution:** 2
**Rating:** 5
**Confidence:** 3

**Summary:**

The authors introduce GAOKAO-Eval, a comprehensive benchmark based on China’s National College Entrance Examination (Gaokao), and conduct closed-book evaluations on LLMs released before Gaokao. This could (partially) address the data leakage issues (only for the models that are released before GAOKAO).

The main contributions of the paper lies on the findings and insights after applying the benchmark on different LLMs. Their findings reveal that even after controlling for data leakage, high scores still fail to truly reflect human-aligned capabilities. The authors introduce the Rasch model from cognitive psychology, and identify two key issues: 1) anomalous consistent performance across various question difficulties, and 2) high variance in performance on questions of similar difficulty.

Finally, the authors recruit human teachers to grade the LLM responses. The grading is inconsistent, and the models show recurring mistake patterns.

The study promotes that reasoning-based approaches like o1 can help mitigate these discrepancies and highlights the need for more LLM-aligned difficulty assessments in future benchmarks.

**Strengths:**

1. A new LLM benchmark with no data leakage is always demanding in the community to have a subjective reflection of LLM performance; however, GAOKAO-Eval itself seems to be only a temporary workaround, as it is likely to be included in the corpus of more recent LLMs.
2. The efforts in the evaluation is non-trivial, including a thorough comparison on multiple LLMs, a new WQX model specialized for GAOKAO task, human grading, etc.
3. The authors reports several interesting findings, including the inconsistency of LLM w.r.t. question difficulty, grading, etc. They also examined the relationship between o1 reasoning tokens and performance consistency. These findings could guide the development of more aligned LLMs.

**Weaknesses:**

1. What does “human-like reasoning” mean? The term is used in several places but lacks a clear definition. More importantly, would “human-like reasoning” still be important if the LLM already achieves “human-like performance”? Addressing these questions could better motivate the research.
2. The performance of the new model is only marginally better than 4o (in “Science Total” and “Art Total”), even after being trained with an extensive GAOKAO-related corpus. What if 4o were fine-tuned on the same (or a subset of the) corpus? Additionally, what is the key message or finding conveyed by including the WQX model in the results? The necessity is unclear.
3. o1’s reasoning ability is mentioned and the finding looks promising; however, the internal reasoning process of o1 is opaque to users and the impact of CoT or other reasoning techniques on white-box models is not explored. Would CoT help reduce the inconsistency?

Minor:
1. line 23: "anomalous consistant performance across various question difficultiess" should be "consistent" and "difficulties".
2. line 25: "we find": "w" should be capitalized.

**Questions:**

1. Would “human-like reasoning” still be important if the LLM already achieves “human-like performance”?
2. What if 4o were fine-tuned on the same (or a subset of the) corpus?
3. What is the key message or finding conveyed by including the WQX model in the results?
4. Would CoT or other reasoning techniques help reduce the inconsistency?
5. After reading through the paper, I still feel unclear about the title: why can’t a high score truly reflect LLM capabilities? If high scores aren’t reliable indicators, how can you conclude that WQX improves over InternLM in the paper based on an increase in accuracy?

**Details Of Ethics Concerns:**

54 high school teachers were involved in grading subjective questions. It is unclear whether the study received IRB approval.

---

### Official Review · Reviewer_BefA · 2024-11-04

**Soundness:** 2
**Presentation:** 1
**Contribution:** 1
**Rating:** 3
**Confidence:** 3

**Summary:**

This paper aims to study if the high scores truly reflect human-aligned capabilities in LLMs. To this end, the authors propose a new eval dataset called GAOKAO-Eval, comprising of different question types, subjects, difficulty levels, etc. Evaluation on this dataset shows that the trained model WQX and GPT4o has much better performance than other models like Qwen, Mixtral, etc. The authors conduct different experiments to show the mismatch between LLM capabilities and the expected human-aligned abilities.

**Strengths:**

Understanding the capabilities of the LLMs is a very relevant and timely topic. I appreciate the author’s effort to curate such a valuable dataset that aims to test various abilities of the models.

**Weaknesses:**

I think the paper can be significantly improved and revised to clearly articulate the experiments, results, and insights.

1.	The paper’s general message that LLMs’ performance varies across similar question types and that there is anomalous consistency across difficulty levels is well-studied in the literature. It would be beneficial if the authors focus on their dataset to showcase how models perform across different subjects and difficulty levels, highlighting what types of problems they perform well on versus those where they fail, and providing potential reasons why. Currently, results are aggregated to show performance variations across models on different difficulty levels.

2.	I found it very difficult to interpret the results, as none of the figures provide a clear explanation of the experiment, insight, or key takeaway. For example, in Fig. 4, you show overall performance across models by subject, but do not clarify what the takeaway is from this figure. Does it imply that WQX and GPT-4o perform the best on this dataset? What is the overall accuracy on this dataset? It’s unclear what the models' performance is on the entire dataset.

3.	Similarly, Fig. 5 lacks an explanation of how human ratings and LLM-based judgments were incorporated into ELO. The graph only shows the difficulty level for 11 questions. What does aligning difficulty level with expert judgments mean? Why are only GPT-4o results shown? What does the difficulty of extracted questions signify?

4.	In Fig. 6, why is the IRT fit across all model results instead of fitting it at each model level to show, for example, whether GPT-4o outputs across difficulty levels align with human abilities? This result is unclear.

5.	Fig. 7a has a grey area—what does this represent? How is difficulty determined by humans or models? The phrase “across models” is also unclear regarding what this graph is meant to demonstrate

6.	In line 357, you mention, “our difficulty ratings are well-aligned with human perception and accurately reflect the human-aligned capabilities of LLMs.” How did you arrive at this conclusion?

7.	What is the takeaway or insight from Fig. 8?

8.	Where is eq 3 applied?

9.	Figure 11 requires more detail. What does incorporating O1 tokens mean? O1 provides the steps and final answer but not the backend exploration or raw tokens, so what is meant by this?

10.	The explanations and insights for Figures 11a, b, c, and d are poorly articulated.

11.	Why not compare with other open-source multimodal models like LLavaOneVision and LLavaNext, which have shown to be more powerful on multimodal data.

12. How were the human raters selected? details of inter-rater agreement etc., should be provided

Apart from the above the key questions for me are:

1.	Given that variants of GAOKAO-Bench and GAOKAO-MM already exist, what is the true novelty of this dataset? While the authors mention it is secure and non-leaky, the other two datasets are as well. What differentiates the construction of this dataset compared to the other two, establishing it as a key contribution?

2.	If the novelty does not lie in the dataset itself, then the key contributions should focus on the insights derived from the data to deepen our understanding of LLM capabilities. Unfortunately, the paper does not fully address this aspect, as the authors primarily report aggregate numbers without clearly presenting key takeaways beyond the general message that LLM performance does not align with human abilities. I would like to see some key insights or takeaways derived from the experiments that are generalizable and hold broader significance for the community.

**Questions:**

see above

---

### Official Review · Reviewer_6gCv · 2024-11-04

**Soundness:** 2
**Presentation:** 3
**Contribution:** 3
**Rating:** 5
**Confidence:** 3

**Summary:**

This paper introduces GAOKAO-Eval, a new benchmark based on China’s 2024 Gaokao exams to assess large language models (LLMs) in a “closed-book” manner, mitigating issues like data leakage. It claims that high scores in existing benchmarks do not necessarily reflect human-aligned capabilities, presenting two main phenomena: “semi difficulty-invariant scoring” and “high variance on similarly difficult questions.” The authors use the Rasch model to analyze scoring patterns and propose “reasoning-as-difficulty” tokens as a potential alignment method.

**Strengths:**

• Introduces a comprehensive evaluation benchmark using Gaokao exams that updates every year with minimal/no data leakage.

• Explores scoring consistency and variance with respect to question difficulty.

• Attempts to model scoring behavior using cognitive psychology (Rasch model).

**Weaknesses:**

• The Rasch model is commonly used in human testing. But it is unclear if the Rasch model is the best fit for modeling LLM behavior, especially without fully exploring/discussing alternative psychometric models.

• Some descriptions seem exaggerated. GAOKAO-Eval primarily assesses knowledge-based aspects of LLM performance, focusing on subject knowledge and question-answering within a constrained exam format. This scope limits its comprehensiveness as a benchmark for LLM capabilities, which is inconsistent with what is described in Section 2 Paragraph 1.

• The process of human involvement is not clear. The study involves 54 human graders without disclosing ethical considerations, which raises potential concerns.

**Questions:**

1. Why was the Rasch model chosen over other psychometric models, and how does it specifically suit LLM evaluation?
2. Can the observed phenomena in GAOKAO-Eval (e.g., high variance in similar-difficulty questions) be verified with non-Gaokao-based tests?

**Details Of Ethics Concerns:**

Details on grader recruitment, data privacy, grader anonymity, workload, and compensation etc. are absent.

---

### Official Review · Reviewer_4Tss · 2024-11-04

**Soundness:** 2
**Presentation:** 2
**Contribution:** 2
**Rating:** 3
**Confidence:** 4

**Summary:**

In order to reveal the limitations of current benchmarks in evaluating human-aligned capabilities, this paper proposes a benchmark based on China’s college entrance exam and conducts evaluations on LLMs released before the benchmark data. The paper finds that LLMs have high variability on questions of similar difficulty and there is performance mismatch between LLMs and human annotators.

**Strengths:**

•	The proposed benchmark highlights the data-leaky issues of previous benchmarks. The annual update of GAOKAO is helpful to evaluate the LLMs performance without tedious manual data collection.
•	The paper evaluates a few popular LLMs on this proposed benchmark.
•	The paper finds that there is a performance mismatch between humans and LLMs when conducting GAOKAO tasks.

**Weaknesses:**

•	The paper lacks clarity:
o	How are the human results conducted? What are the grading guidelines? How to distribute the tasks? How to validate the human evaluation process?
o	The paper uses Rasch model to simulate human performance. However, there lacks clarifications why GAOKAO performance could be simulated by Rasch model. The actual human performance distribution might be similar to the LLM’s.
o	Line 274 mentions the difficulty of questions. How is exactly the hybrid approach with human annotations and LLM scores?
•	The paper claims that o1’s reasoning-as-difficulties can mitigate the mismatch between the human performance and LLM’s performance on the benchmark. However, the paper lacks experiments on the performance distribution of o1 on the benchmark, and it is still unknown how this performance distribution aligns with the actual human performance distribution, which is also lacking in the paper.
•	The paper contains a few grammar errors and typos: Line 23: ‘consistant’, ‘difficultiess’, Line 26: ‘we’, ‘phenomenon’ should be plural, Line 459: ‘cabilities’, Line 527: ‘discoverys’, and more.
•	The motivation of this paper is questionable. Previous benchmarks such as GAOKAO-MM and GAOKAO-Bench are proposed to evaluate the comprehensive capabilities of LLMs. However, this paper shows another point that human-aligned LLMs should have similar performance distribution as humans. Wouldn’t LLM research make better LLMs that have higher scores on tasks where humans perform poorly? Unlike improving safety and reducing toxicity through human-alignment, aligning human capability in reasoning tasks might not be a good idea.

**Questions:**

Would you please address the concerns in weakness part?

---

### Meta-Review · Area_Chair_Pd1Z · 2024-12-17

**Metareview:**

This paper proposes a benchmark, called GAOKAO-Eval, based on China’s 2024 National College Entrance Examination for the evaluation of large language models (LLMs) in a “closed-book” setting. Based on the study, it concludes that LLMs receiving high scores in existing benchmarks do not necessarily reflect human-aligned capabilities.

Major strengths:
- Evaluation of LLMs is an important topic to study.
- The proposed benchmark nicely addresses the data leakage problem in existing benchmarks.

Major weaknesses:
- The motivation of this work needs to be articulated better.
- Clarity of the presentation has room for improvement.

The authors should be praised for making this attempt to address an important topic, but the motivation underlying the study and the design of the methodology need better articulation to make this work and its findings more convincing. The authors are encouraged to improve their paper for future submission by considering the comments and suggestions of the reviewers.

**Additional Comments On Reviewer Discussion:**

The authors did not respond to the reviews.

---

### Decision · Program_Chairs · 2025-01-22

Reject